# Primary Education Truancy and School Performance in Social Exclusion Settings: The Case of Students in Cañada Real Galiana

**Santa Lázaro** [1,*]**, Belén Urosa** [2] **, Rosalía Mota** [1] **and Eva Rubio** [1]

1 Department of Sociology and Social Work, Faculty of Humanities and Social Sciences, Comillas University, 28049 Madrid, Spain; rmota@comillas.edu (R.M.); erubio@comillas.edu (E.R.)

2 Department of Education, Research Methods and Assessment, Faculty of Humanities and Social Sciences, Comillas University, 28049 Madrid, Spain; burosa@comillas.edu

* Correspondence: slazaro@comillas.edu

**Abstract:** Academic studies show that one of the main predictors of early school dropout at secondary education is student truancy behaviour, usually beginning at primary education. This is a problem that gets worse in socially vulnerable environments. This study analyses the prevalence and types of truancy in a population of students with high social risk in Madrid city, studying the relationship between truancy and their school performance. A questionnaire was answered by mentor teachers of students at the preschool and primary stages (N = 120), who reported information from a total of 433 students from 12 different schools. Results showed a high level of prevalence in the different types of truancy (Active and Passive). Among these behaviours, 46.86% of students skipped entire days without a valid excuse and 42.51% did not usually do their homework. Overall, 60% showed underachievement and great rates of curricular gap. In 6th grade, the last year of primary school, 74.42% of students had repeated one grade and 27.91% more than one. Moreover, significant correlations were found between truancy and school performance. The detection and early action against truancy in primary education with This type of student will reduce early dropouts and make school a key actor for the development of these students.

**Keywords:** truancy; dropout; school performance; child poverty; socially vulnerable students; preschool; primary school

## 1. Introduction

The importance of education as a tool for proper individual development is indisputable. This is especially true when it comes to people living in situations of poverty and social vulnerability, which are often passed down through generations and "trap" individuals in a spiral that is difficult to get out of. In these specific contexts, school is considered to be the setting for learning and coexistence, becoming a critical factor for overcoming the inertia in the cycle of poverty and providing strength to facilitate integral development.

Nonetheless, more than 262 million children and young adults worldwide do not attend school. Moreover, six out of 10 children have not yet acquired basic literacy and arithmetic skills after several years of study.

Education is a human right and an essential element to achieve sustainable development. This is reflected in the United Nations 2030 Agenda, known as Sustainable Development Goals, and formally adopted in September 2015 by all its Member States. Goal 4 of the Agenda aims to "ensure inclusive and equitable quality education and promote lifelong learning opportunities for all". Its Target 4.5 is specially directed to "eliminate gender disparities in education and ensure equal access

at all levels of education and vocational training for the vulnerable, including persons with disabilities, indigenous people and children in vulnerable situations".

In order for education to be a "social elevator" and a tool for the achievement of adequate progress and social equity, it is important for students to remain within the education system throughout its different stages (from early childhood education, primary education, compulsory secondary education, up to high school or vocational training), actively participating in school life and acquiring the skills required for adulthood.

Disconnecting from or leaving the education system without completing all these stages is known as early school dropout, representing the percentage of the population between 18 and 24 years of age who have not completed post compulsory secondary education and are no longer receiving any type of education or training.

Within the European Union, Spain has the highest level of early school dropout, reaching 21.4% in 2019 [1]. In the Community of Madrid, the 2019 level was lower at 12.4% [2].

One of the most important predictive factors for school dropout is the student's dissociation or disconnection from school and his or her lack of motivation when it comes to life opportunities [3–8]. This "uncoupling" process does not happen suddenly during the last stages of compulsory education, but rather gradually since primary education [9,10]. As a result, the truant students show poor school performance and move through the school years accumulating knowledge gaps or having to repeat years, losing any relationships they have established. For these students, their school is far from being a source of support and personal affirmation.

It is important to consider that school truancy does not only involve not attending school [11], but also occasionally, disguised by more or less normal attendance, there can be underlying emotional and motivational distancing processes that are more likely to lead to early school dropout [3,12].

In Spain, school is compulsory between the ages of 6 and 16, which is why the prevalence of truancy is low, especially at the primary education stage. However, the rate is higher amongst children from families facing poverty and social exclusion, even whilst regular attendance is usually an essential condition for receiving benefits and social support, as is the case with the government of Madrid's guaranteed minimum income. This compulsory nature of attending school can lead to truancy manifesting itself in different forms.

## 1.1. The Concept and Types of School Truancy

One of the primary obstacles in the study of school truancy is its conceptual delimitation. The difficulty lies in the fact that it is a complex phenomenon that can manifest itself in different ways, each of which can have different implications and consequences, which is why the focus of the study varies based on the dimensions that are prioritised [13]. The concept has different nuances depending on the approach, whether it be from the point of view of the student, the family, the school, or a social and cultural perspective. Therefore, as indicated by Rué [14], in order to understand the phenomenon of truancy, a comprehensive view must be taken that incorporates sociological, psychological, and curricular aspects, whilst also hearing from those people who are truant and the reasons behind this.

Most of truancy literature shows the negative effect of This phenomenon on the school performance [15–23].

The consequences are not only academic but also affect other psychological and social spheres of the student's life. The consequences of This phenomenon affect different aspects of student life: academic, social, and psychological [24–26], as not only does truancy affect their school performance, but it also affects the main aspects of life such as their self-concept and perception of self-efficacy, social relationships, the sense of belonging to a community, and, in the long run, employment opportunities and social promotion. These consequences become even worse amongst children whose families live in poverty, as they do not have the resources and necessary skills to cushion the blow and compensate for the shortfalls that result from truancy [11,15].

From an administrative and regulatory point of view, school truancy is defined as "when a student does not attend school without a good excuse, if they are in the compulsory education age group" [27]. This definition considers aspects such as frequency, severity, and timescale. Based on these, truancy can have different types and levels (Table 1) that can range from mild to not attending school at all; these are severe truancy (16–30 days or specific periods/term); moderate truancy (absent between 10–15 full days or specific periods/term); minor or infrequent truancy (absent fewer than 10 full days or specific periods/term); tardiness (attending school, but arriving late); or virtual or Passive Truancy, in which the student attends lessons but is absent from an emotional or motivational point of view [28,29]. This last type is also known as "inner" truancy or "present but absent", referring to students who display off-task behaviours and do not get involved in school activity, wasting time and accumulating educational problems throughout the school years [25].

**Table 1.** Types of Truancy.

| Type of Truancy | Associated Behaviour |
| --- | --- |
| No attendance | Not attending school at all |
| Severe | Absent between 16–30 days or specific periods/term |
| Moderate | Absent between 10–15 full days or specific periods/term |
| Minor or infrequent | Absent fewer than 10 full days or specific periods/term |
| Tardiness | Attending school, but arriving late |
| Virtual or passive | Attending lessons but absent from an emotional or motivational point of view |

Source: own elaboration, adapted from Martínez [29].

Other classifications distinguish between severe truancy (when absence is over 50% of school days), average truancy (when absence is between 25% and 50% of school days), and minor truancy (when absence is less than 25%). In Madrid, the truancy intervention protocols are activated when a student misses five days per month in primary education, and 25 school hours or 25% of school days during compulsory secondary education [30,31].

*1.2. Prevalence of Truancy in Madrid*

As of 1 January 2020, Madrid had a total population of 3,334,730 [32], of which 495,363 were under the age of 16. Of these, 208,703 were aged between 6 and 12, which is the age group that corresponds to primary education in Spain [33]. In terms of education data, in the 2018–2019 academic year, 287,810 children were in preschool education, 432,889 were in primary education, and 280,313 were in compulsory secondary education in Madrid.

In order to understand the data behind the prevalence of school truancy in Madrid, it is necessary to refer to the School Truancy Prevention and Control Programme. This programme has been in place since 2001 with the aim of facilitating the prevention and control of school truancy amongst students in primary education and compulsory secondary education (ages comprising 6–16), whilst only having a preventive approach for preschool (3–6 years). The School Truancy Prevention and Control Programme is implemented throughout the school year, conducting customised socio-educational interventions adapted to the needs of each student and their families, in coordination with the schools, municipal services, and NGOs. During the 2018–2019 academic year, the School Truancy Prevention and Control Programme dealt with 4025 children within the compulsory education age group, that is, both primary education and compulsory secondary education. In terms of the split by gender, 51.18% were male and 48.82% were female.

The distribution of truant students varies across the city's 21 districts, which are administrative and territorial divisions of Madrid. The highest levels are found in the southern area of the city: Puente de Vallecas (364 cases), Villa de Vallecas (211 cases), Carabanchel (356 cases), Villaverde (240 cases), Usera (213 cases), and La Latina (206 cases). These districts are indeed those with the highest Analytic

Hierarchy Process (AHP) Vulnerability Index in Madrid. The City Council of Madrid uses This indicator to evaluate vulnerability and risk of social exclusion in the distinct areas of the city [34].

With regards to the type of school, the students mainly attend public schools. At primary education level, 1457 students were dealt with at public preschools and public primary schools, 355 students at publicly funded private schools, and just six at private schools.

### 1.3. School Truancy in Poverty and Exclusion Contexts

School truancy is more prevalent in settings where there is poverty and social exclusion, and, at the same time, becomes a factor that can lead to social marginalisation [6,35,36], given that the truant children stop receiving the essential learning required for their future. Studies have shown the link between school truancy and other factors, such as the socioeconomic context or family environment [10,24,37,38]. It can be said that there is an intergenerational transmission of poverty as people who grow up in needy families and low-income households are more likely to suffer from financial problems and impoverishment in their adult lives [35,39]. In these cases, as well as a poverty gap there is an educational gap, with children's school attendance being more precarious and less regular.

This is the case amongst some of the children who live in Cañada Real Galiana in Madrid. This is one of the largest settlements of substandard housing in Europe and covers an area of 14.2 km in the Community of Madrid. It is split into six sectors and spreads across the municipalities of Coslada, Rivas Vaciamadrid, and the capital city of Madrid. According to the latest census, published in 2011, and the Ombudsman for Children's report of the same year, 5666 people live in Cañada Real Galiana, 42.24% of which are below the age of 18. As data show, This settlement is mainly inhabited by a young population, with an average age of 25.1 [40]. Whilst 16% are under the age of 5, 38% are teenagers, 35% are aged between 20 and 40, and only 4.3% are over the age of 65. Sectors five and six, which constitute the territorial area under research, belong to Vallecas Villa and Vicálvaro districts. These sectors have a population of 4554 people and 1812 children under the age of 18. It is important to consider the difficulties associated with having updated data on This population, since there is no administrative registry compiling this information.

The population of Cañada Real Galiana is demographically diverse, both in terms of the residents' country of origin (33% are foreigners) as well as in their cultural-ethnic group (there is a large population of Spanish and foreign Roma people). In total, 38.7% are Spanish Roma, 27.2% are Spaniards (not Roma), 26.1% are of Maghrebi origin (mainly from Morocco), and 7.8% are foreigners of other nationalities. This diversity increases the social rejection that affects This population, as the most rejected groups in Spain are people of Roma origin and the Muslim population [41], which leads to high levels of social vulnerability. As a result, Cortes, Morenos and Andújar [40] indicate that 12% of This population have extreme vulnerability indicators and 28.5% have a high or average indicator, which means that approximately 40% of the population are vulnerable. A total of 48% have low vulnerability indicators, leaving just 11% who fall outside of these levels.

The lack of community resources is one of the key characteristics of Cañada Real Galiana, including educational resources, sanitary resources, the lack of public spaces for leisure activities such as sports centres or cultural centres, and other public, private and publicly funded private resources of a formal nature. Lack of infrastructure limits the equality of opportunities for accessing education [42,43]. By not having the necessary resources in their immediate environment, children have to leave the settlement to attend their schools, which are far removed geographically. Providing children's transport alternatives, school bus routes have been set up specifically for students in the compulsory stages (between the ages of 6 and 16), given the limitations of public transport and poor road quality. The pickup times on the school bus mean that children are unable to take part in activities outside of the school schedule, whether these are extracurricular or trips, excursions, or cultural visits. This accessibility issue is worse during the non-compulsory stages of education, as there are no bus routes at all.

In addition to the difficulties of accessing education, a social programme evaluates the situation in Cañada Real: the Intercultural Community Intervention Project [42]. This programme, a social programme developed in Cañada Real, highlights the situation regarding school truancy and the experiences of the children of Cañada Real Galiana when behind the normal grade level for their age. Moreover, it also highlights the difficulties within the process of promotion from one stage of education to another (from primary to secondary education), as well as situations in which some family members lack involvement in their children's education. These problems do not differ from those frequently experienced by children of Roma ethnicity from other geographic areas [10,24,44], that is, a lack of continuity and educational success, especially during compulsory secondary education and the other non-compulsory stages [45].

Although the right to education has been guaranteed for many years, and there has been clear progress with the rate of school attendance among Roma children at preschool and primary school, it is important to emphasise that there is a significant educational gap between the Roma population and the rest of the population. According to the indicators established in the National Roma Integration Strategy in Spain 2012–2020 and its intermediate evaluation, which took place in 2016, there are high levels of school dropout at the secondary school level: 61% of Roma males and 64.3% of females aged between 12 and 24 [46], compared to 20% of the overall population aged between 18 and 24. Furthermore, there is a very low presence of Roma people in post compulsory education, with 8% of the young Roma population not reaching This level, up to the age of 25.

Of particular concern is the level of school abandonment amongst Roma girls during the early education stages, between the ages of 10 and 14 [44]. It is important to remember that young Roma girls not only have to fight against gender prejudice, but also against cultural and social prejudices, facing many more educational barriers [47], which do not only stem from the traditional role they carry out within their families, but also from the lack of educational expectations placed on them by their schools. These low expectations and negative stereotypes still present in society can significantly impact Roma children, increasing their sense of loneliness as a result of not having Roma role models in the higher levels of the education system [45].

In the face of these challenges, different socio-educational programmes have been implemented in Cañada Real Galiana by socio-educational institutions and non-governmental organisations. They carry out primarily educational support and reinforcement programmes to complement formal education, as well as informal education activities based on leisure activities, aimed at developing personal and relationship skills. This socio-educational activity is part of the different participation areas in Cañada Real and has a long and recognised history, having helped to create collaborative ties and strengthen networks with the support and participation of public and private entities [48].

With their activities, these entities seek to promote educational success, understanding that This does not just involve obtaining academic results, but also acquiring new skills, competences, and motivation, all of which demonstrate personal maturity and prepare the children for the future.

As a result, it is necessary to study truancy in primary education because, as indicated by Hernández and Alcaraz [49], the early identification of risk factors relating to early educational abandonment begins in primary education and is critical for its prevention. It is important to be able to identify the risk factors that cause early educational abandonment in order to design effective prevention and intervention proposals [15,19].

This study involves a diagnostic approach to the phenomenon of school truancy in Cañada Real Galiana that can support the design and implementation of future educational initiatives. As a result, the prevalence of school truancy has been examined amongst preschool and primary school students in Cañada Real, analysing the different types and profiles. The relationship between level of truancy and school performance has been also investigated. It is important to highlight that the study tackles a research topic that is not frequently covered in papers on truancy, as these usually focus on its prevalence within secondary education and not on early school truancy.

## 2. Materials and Methods

The research presented has an exploratory and descriptive nature and was conducted through a survey that collected information from Cañada Real's enrolled students.

### 2.1. Participants

The data on children's truancy levels and their academic results were collected through the information provided by the mentor teachers of children in preschool and primary education during the school year 2018–2019. At these educational levels, mentor teachers are the ones in charge of working all the school subjects with the children, becoming, as well, the reference figures of reference for the rest of the educational team and the families.

Mentor teachers were chosen as participants for different reasons. On the one hand, despite the fact that there are census data based on the neighbourhoods and district registries, it is difficult to confirm their correspondence with the real location of the population in the given area. Therefore, they were key for having a reliable data sample frame to later construct a random sample of boys and girls. On the other hand, the study examines a population in a highly vulnerable context, which is difficult to access, especially in the case of minors, whose participation must have the informed consent of their parents or legal guardians.

In order to have the sample as representative as possible, all the preschools and primary schools with students who lived in Cañada Real Galiana in the Villa de Vallecas and Vicálvaro geographical areas were contacted, which added a total of 21 educational centres, with 696 boys and girls in preschool and primary education. The list of schools and data on students enrolled were provided by the Education Department of Madrid's government.

A total of 12 out of the 21 contacted schools responded to the survey, 10 public and two publicly funded private schools. Altogether, 120 mentor teachers reported information from a total of 433 children. The group of children from whom information was obtained represented 62.21% of the children of Cañada Real Galiana enrolled in preschool and primary education in the two districts considered. It is not possible to provide information on the specific mentor teachers' response rate, since the list of schools' mentor teachers was restricted to use by the centres themselves and the Education Department of Madrid's government.

Table 2 shows the list of schools, ranked by their relative proportion of students from Cañada, and the distribution of students by grade.

Table 3 shows the total number and distribution by school of mentor teachers who participated and its relative weight in the sample. It includes, as well, information on the total number and distribution of students whose information was obtained, their relative weight in the sample, and the percentage they represent over the total number of children enrolled.

In terms of the representation from preschool and primary school levels, mentor teachers at all grades of preschool and primary school years responded to the questionnaire (Table 4).

**Table 2.** Schools with students enrolled who lived in Cañada Real Galiana, Villa de Vallecas and Vicálvaro districts. 2018–2019 school year.

| School | % Students Cañada | % Preschool | % Primary |
|---|---|---|---|
| Ciudad de Valencia | 23.99 | 17.97 | 25.49 |
| Blas de Otero | 17.67 | 24.22 | 16.28 |
| Honduras | 11.64 | 11.72 | 11.15 |
| Doctor Severo Ochoa | 11.35 | 7.81 | 12.21 |
| Francisco Fatou | 6.90 | 6.25 | 7.08 |
| Alfonso X El Sabio | 5.75 | 9.38 | 4.96 |
| Nueva Castilla | 4.31 | 0 | 5.31 |
| Torrevilano | 4.17 | 5.47 | 3.89 |
| Agustín Rodríguez Sahagún | 3.02 | 4.69 | 2.65 |
| Juan Gris | 2.01 | 2.34 | 1.95 |
| Loyola de Palacio | 1.87 | 0.78 | 2.12 |
| Sagrado Corazón | 1.29 | 0.78 | 1.42 |
| Vicálvaro | 1.15 | 3.13 | 0.71 |
| José Echegaray | 1.01 | 0.78 | 1.06 |
| Zazuar | 1.01 | 0 | 1.24 |
| Gredos San Diego Las Suertes | 0.72 | 1.56 | 0.53 |
| Pedro Duque | 0.57 | 1.56 | 0.35 |
| Liceo Versalles | 0.57 | 0.78 | 0.53 |
| San Eulogio | 0.57 | 0.78 | 0.53 |
| El Quijote | 0.29 | 0 | 0.35 |
| Carmen Laforet | 0.14 | 0 | 0.18 |
| TOTAL | 100 (N = 696) | 100 (N = 128) | 100 (N = 565) |

Source: own elaboration.

**Table 3.** Number and distribution of participating mentor teachers and students about whom information was obtained. 2018–2019 school year.

| School | N Mentor Teachers | % Mentor Teachers | N Students | % Students | % Students Cañada |
|---|---|---|---|---|---|
| Ciudad de Valencia | 35 | 29.17 | 160 | 36.95 | 22.99 |
| Blas de Otero | 16 | 13.33 | 92 | 21.25 | 13.22 |
| Honduras | 7 | 5.83 | 31 | 7.16 | 4.45 |
| Doctor Severo Ochoa | 14 | 11.67 | 68 | 17.70 | 9.77 |
| Francisco Fatou | 4 | 3.33 | 13 | 3.00 | 1.87 |
| Alfonso X El Sabio | 6 | 5.00 | 16 | 3.70 | 2.30 |
| Agustín Rodríguez Sahagún | 11 | 9.17 | 14 | 3.23 | 2.01 |
| Juan Gris | 4 | 3.33 | 6 | 1.39 | 0.86 |
| Loyola de Palacio | 9 | 7.50 | 18 | 4.16 | 2.59 |
| Vicálvaro | 8 | 6.67 | 9 | 2.08 | 1.29 |
| Zazuar | 1 | 0.83 | 1 | 0.23 | 0.14 |
| Gredos San Diego Las Suertes | 5 | 4.17 | 5 | 1.15 | 0.72 |
| TOTAL | 120 | 100 (N = 120) | 433 | 100 (N = 433) | 62.21 (N = 696) |

Source: own elaboration.

**Table 4.** Mentor teachers whose information was recorded. 2018–2019 school year.

| Grades | Mentor Teachers | % |
|---|---|---|
| 1st grade of primary school | 13 | 10.83 |
| 2nd grade of primary school | 18 | 15.00 |
| 3rd grade of primary school | 13 | 10.83 |
| 4th grade of primary school | 15 | 12.50 |
| 5th grade of primary school | 16 | 13.33 |
| 6th grade of primary school | 17 | 14.17 |
| Preschool Age 3 | 6 | 5.00 |
| Preschool Age 4 | 12 | 10.00 |
| Preschool Age 5 | 10 | 8.33 |
| **TOTAL** | **120** | **100.0** |

Source: own elaboration.

### 2.2. Variables and Instrument

In order to study school truancy in the preschool and primary school stages, variables were selected that helped to frame the existing problem: the prevalence of different types of truancy, the children's family characteristics, and their academic results. Although there were no precedents in literature, we proposed classifications of truancy that we called Active Truancy, which referred to those behaviours of physical absence from school or classes, and Passive Truancy, in which students attended lessons but displayed off-task behaviours and did not get involved in school activity.

Table 5 shows these overarching variables and their disaggregation into specific variables.

**Table 5.** Research variables.

| Themes | Variables | |
|---|---|---|
| Prevalence of the different types of school truancy | Active Truancy | <ul><li>Skipping whole days.</li><li>Skipping half a day.</li><li>Being at school but not attending lesson.</li><li>Arriving late in the morning.</li></ul> |
| | Passive Truancy | <ul><li>Not paying attention and switching off from activities and explanations.</li><li>Not doing homework.</li><li>Not bringing sourcebooks or other school materials.</li></ul> |
| Families of the truant children | <ul><li>The family's ethnic group.</li><li>Large family.</li><li>Family with support network.</li><li>Family without coexistence problems.</li><li>Single parent families.</li><li>Families with financial problems.</li><li>Families with social risk behaviour.</li><li>Broken families.</li></ul> | |
| School performance | <ul><li>Performed below-average academic performance.</li><li>Repeated at least one grade.</li><li>Repeated more than one grade.</li><li>Failed more than four subjects.</li><li>Failed due to their absences.</li><li>Displayed learning problems.</li><li>Had sensory, motor, or cognitive disability.</li><li>Had no study skills.</li></ul> | |

Source: own elaboration.

*2.3. Procedure*

By using EUSURVEY, a questionnaire was created for mentor teachers. This online survey management tool was developed by the European Commission, which ensured the necessary data protection measures were in place. In order to maintain the confidentiality of the data, no specific information was collected relating to individual students. Instead, anonymous data of the overall group of students were collected for both preschool and primary school.

The questionnaire was sent to the head teachers of all the centres, along with a cover letter and two documents: information and support letters for the study from the Technical Director of the Community of Madrid's Commission for Cañada Real and from the City Council of Madrid's Commission for Cañada Real. Both Commissions are the bodies that coordinate the policies of the municipal and regional governments in Cañada Real Galiana. In addition, they serve as an interface between the different administrations involved and the non-governmental organisations that carry out educational and social programmes in the area. Contact with the mentor teachers was facilitated by each school's head teachers, who were asked to send the mentor teachers a link to the questionnaire.

Collection of the questionnaires completed by mentor teachers started in January 2019 and took three months. The collected data relate to the 2018/2019 academic year.

*2.4. Data Analysis*

Descriptive analyses were conducted on the truancy prevalence, family characteristics, and educational outcomes of children as reported by participating mentor teachers. To carry out these analyses, the direct scores provided by their answers were transformed according to the weight represented by the number of students each mentor teacher had. In addition, several correlation analyses were conducted between (1) Active and Passive Truancy variables and truancy and family characteristics, and (2) truancy and school performance. The statistic used was Pearson's r. The analyses were performed using version 26 of the SPSS programme. The research data can be found in the Supplementary Materials.

**3. Results**

*3.1. The Prevalence and Different Types of School Truancy*

In terms of prevalence, it is important to highlight that not all children who live in Cañada Real Galiana displayed school truancy behaviours. The mentor teachers from each class indicated that 80.43% of students were punctual and more than half (52.17%) did not miss any lessons, or if they had, they had a valid excuse.

However, there were children affected by school truancy. More than three quarters (76.9%) of the mentor teachers who participated stated there were students in their classrooms who skipped entire days without a valid excuse. Almost a quarter (23.3%) indicated that some children skipped the afternoons, having attended in the mornings, and 15% said there were children in their classrooms who arrived late in the morning (Figure 1). The other physical absences (going to school but missing lessons or skipping certain subjects) were hardly mentioned by mentor teachers.

Passive Truancy was more widespread in all its manifestations (Figure 2). More than half of mentor teachers stated that they had students in their classrooms who did not pay attention and get distracted (62.4%), and who regularly did not do their homework (60.7%), whilst 45.3% taught children who regularly did not bring their sourcebooks or other school materials.

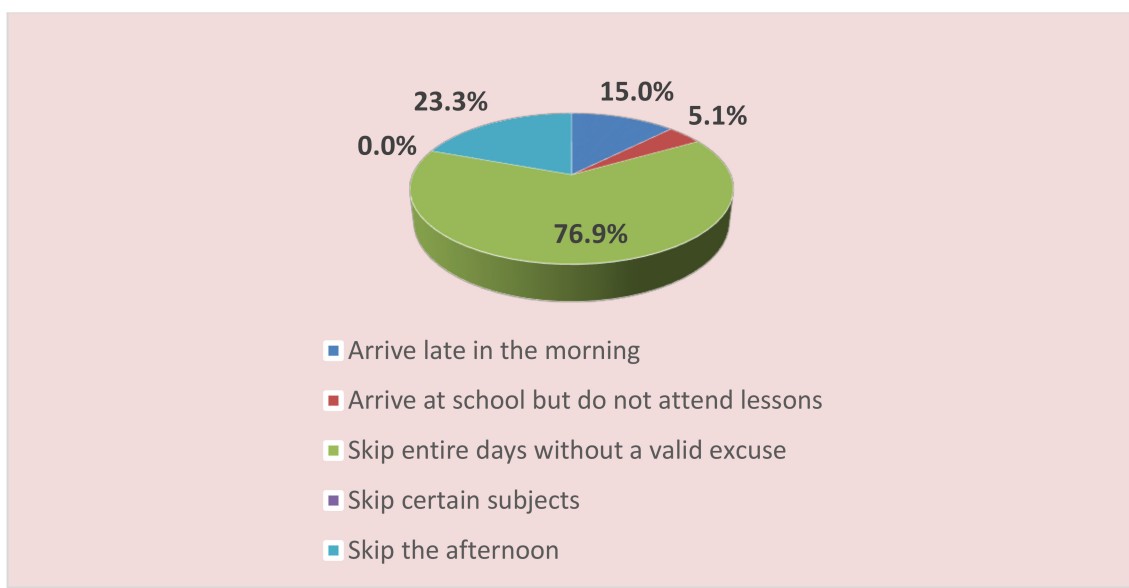

**Figure 1.** Mentor teachers with Actively Truant students by type.

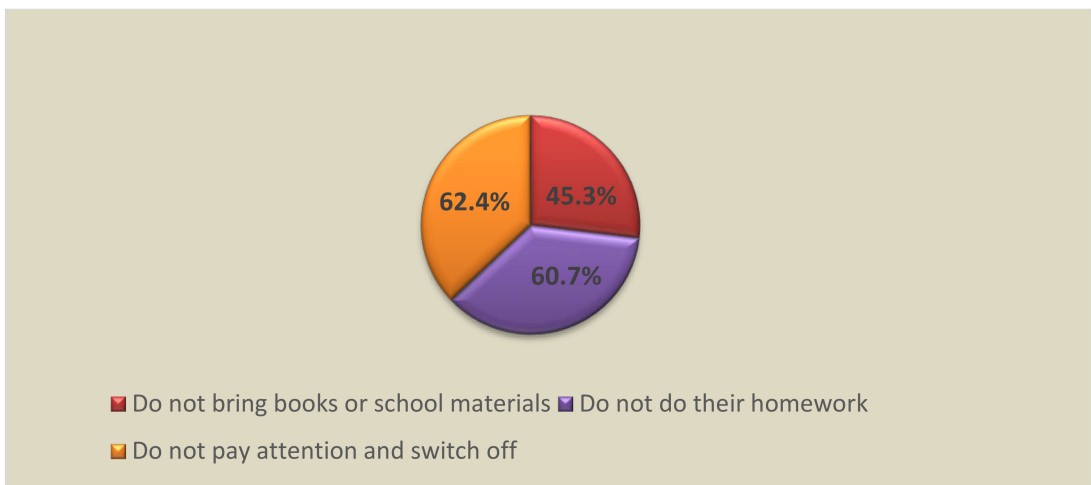

**Figure 2.** Mentor teachers with Passively Truant students by type.

The prevalence of the different types of truancy is shown in Figure 3.

Within Active Truancy, skipping full days without a valid excuse was the most common truancy behaviour amongst children, representing 46.86 percentage of the total. The other Active Truancy situations were much less common. Skipping school on some afternoons represented 9.42% of the identified behaviours, arriving late in the mornings stood at 6.76%, and being at school but not attending lessons represented 4.35% of the total. None of the children from Cañada skipped only some subjects.

In terms of Passive Truancy, the most common phenomenon included students not doing their homework (42.51%), This type of Passive Truancy being at similar levels as the traditional measure of truancy: unjustified absences from school. With regards to the other Passive Truancy behaviours, mentor teachers indicated that 37.88% of children from Cañada did not pay attention and got distracted during lessons, whilst almost a quarter of the total interviewed students went to school without sourcebooks or other required materials (23.91%). It is important to highlight how extremely common Passive Truancy was.

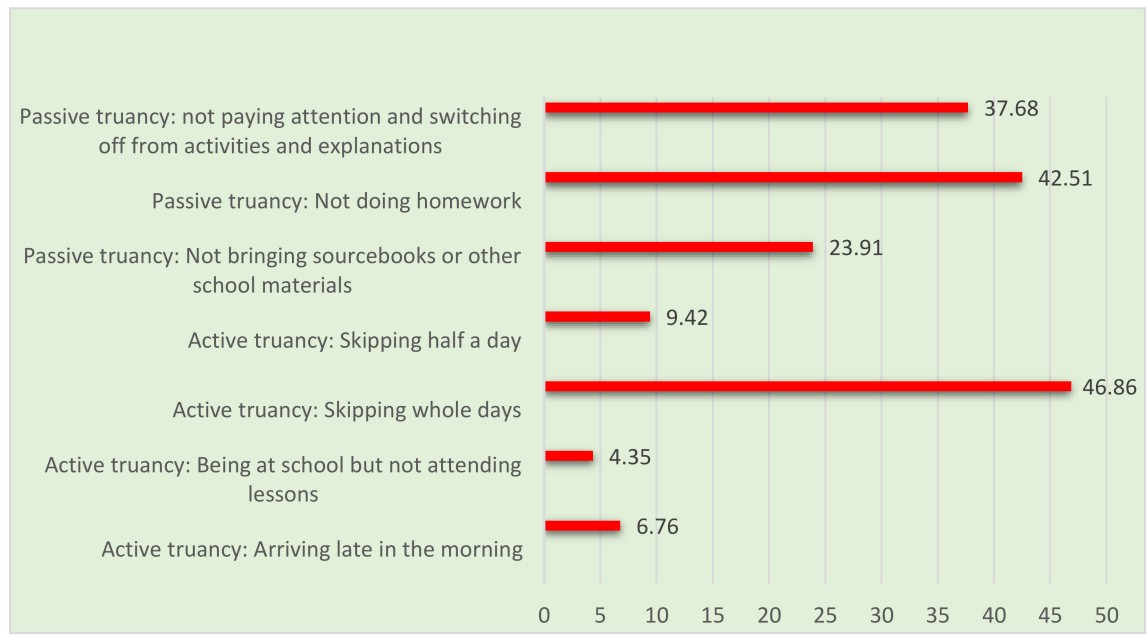

**Figure 3.** Types of truancy (%).

The following figures on Active Truancy show the distribution by grade of preschool and primary school for each type of Active Truancy. Firstly, it is important to highlight the greater prevalence of physical absence at the primary education stage, as shown in Figures 4–7. However, skipping entire days in preschool was of particular relevance and similar to the primary school levels (Figure 4).

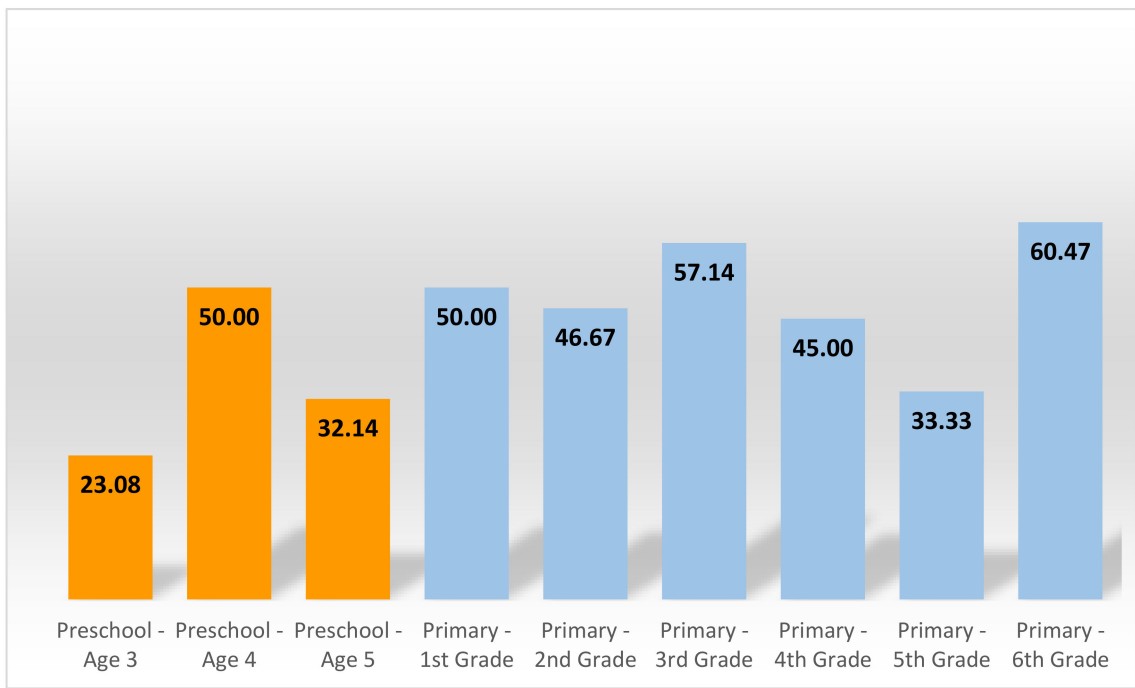

**Figure 4.** Students who skipped entire days without a valid excuse (%).

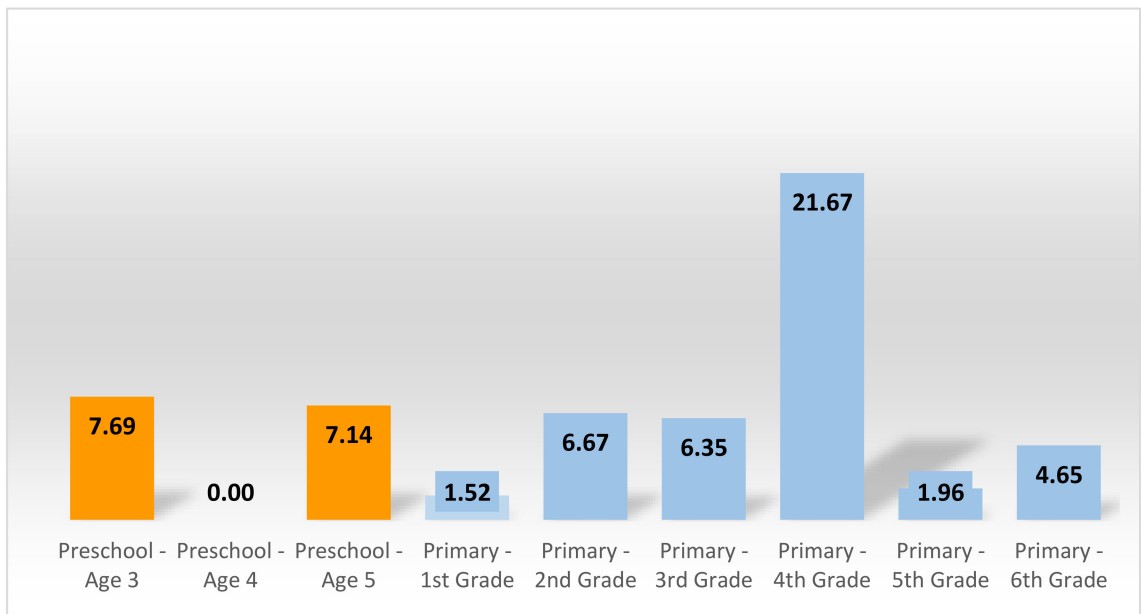

**Figure 5.** Students who arrived late in the morning (%).

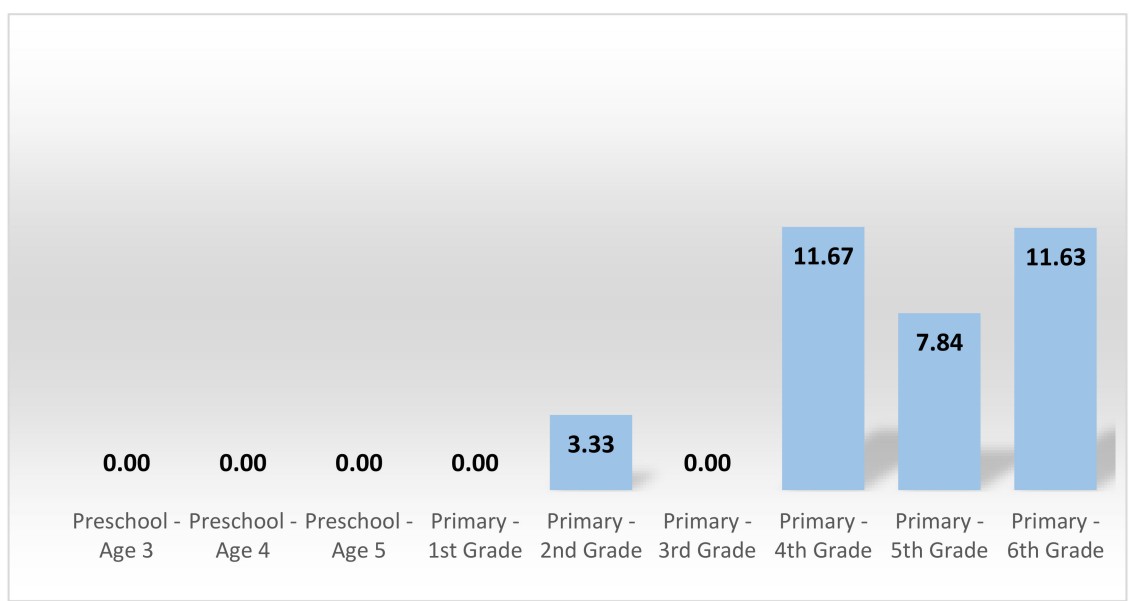

**Figure 6.** Students who went to school but did not attend lessons (%).

The relative weight of preschool students who arrived late in the morning was also significant, as it was higher than all primary school grades except 4th grade (Figure 5).

Furthermore, it was clear that unexcused absence from lessons throughout the whole school day was consistent across all school grades, especially in the primary stage (Figure 4). However, other types of Active Truancy had greater relevance in specific years: the relative weight of the group of students who skipped entire days in the 2nd year of preschool (50%), the students who arrived late in 4th grade (21.67%), those who skipped some lessons in the afternoons in 1st and 3rd grade (approximately 14% in both cases), and, finally, students who were at school but did not get involved in lessons in the last three years of primary school (Figure 6)

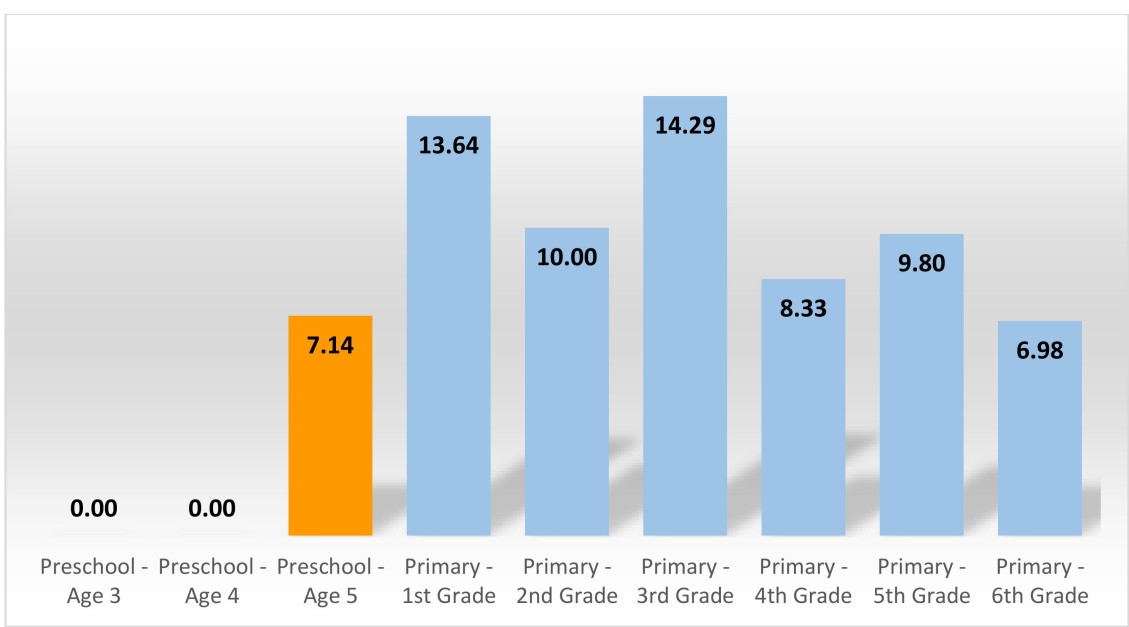

**Figure 7.** Students who were not at lessons some afternoons (%).

Lastly, it is important to emphasise that the 3rd and/or 4th grades of primary school were the years with a greater prevalence of Active Truancy (Figures 4–7), whilst the last year of primary school also ranked highly in terms of entire days skipped. According to the teachers who participated in the study, six out of every 10 children from Cañada Real Galiana had This behaviour (Figure 4).

Regarding Passive Truancy by stage and grade, the study showed that, in certain school grades, This type of truancy reached levels of prevalence similar to the unjustified physical absences. Not completing homework was the most common type of Passive Truancy, both in preschool and primary school, as shown in Figure 8. More than 30% of these students did not do their homework regularly at primary school, increasing to 50% amongst 1st and 4th grade students at This education level.

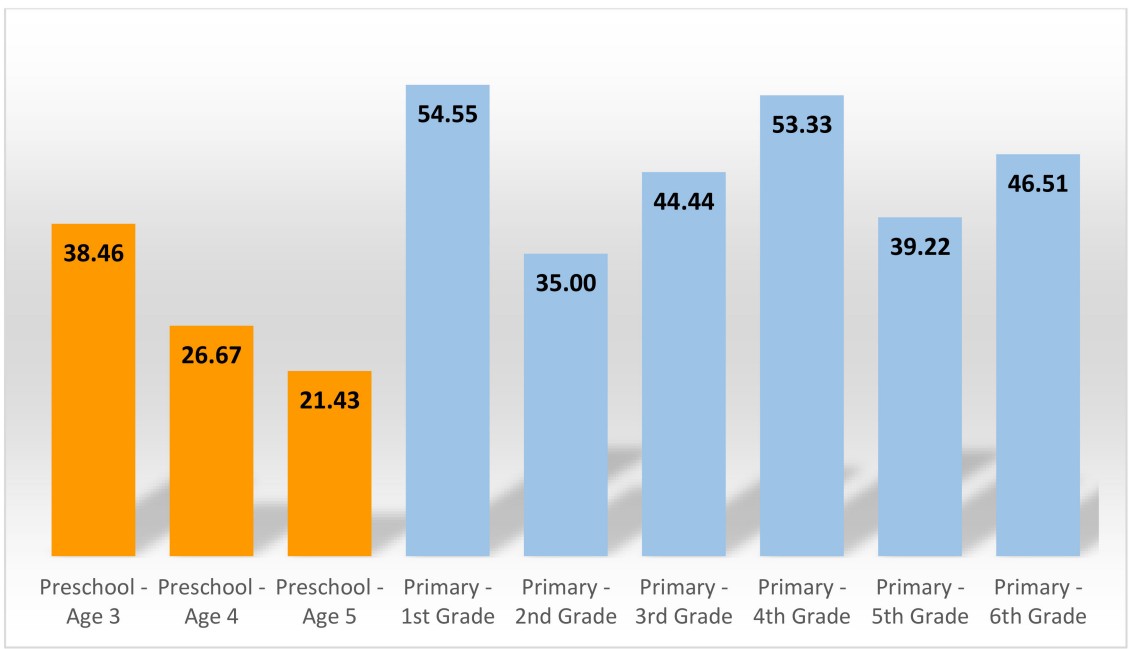

**Figure 8.** Students who did not do their homework (%).

The number of students who got distracted during lessons and did not get involved in class time activities was also noticeable (Figure 9), with at least a third of students displaying This behaviour throughout each year of primary school. Attending classes without sourcebooks and other school materials was less common (Figure 10). Although it was during primary school where Passive Truancy affected a greater number of students, its incidence was not insignificant during preschool grades, where between 20% and 30% of children did not do their homework, did not attend classes, or did not take school materials with them. This trend did not increase at the higher school grades in either of the stages, as the prevalence changed every grade.

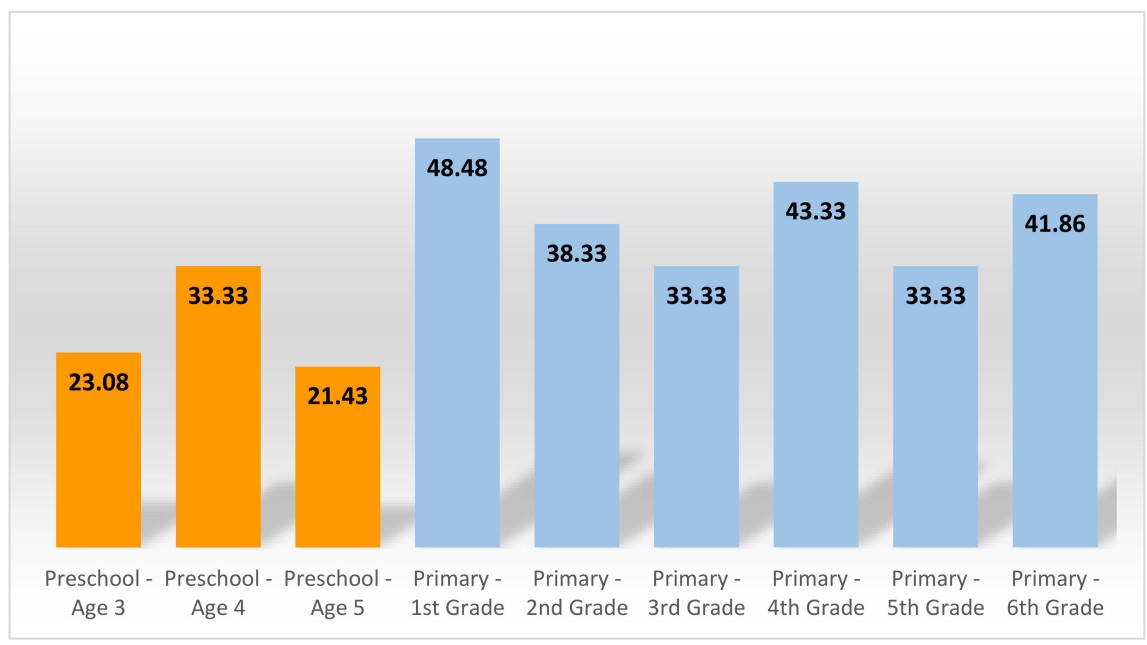

**Figure 9.** Students who did not pay attention or switched off during lessons (%).

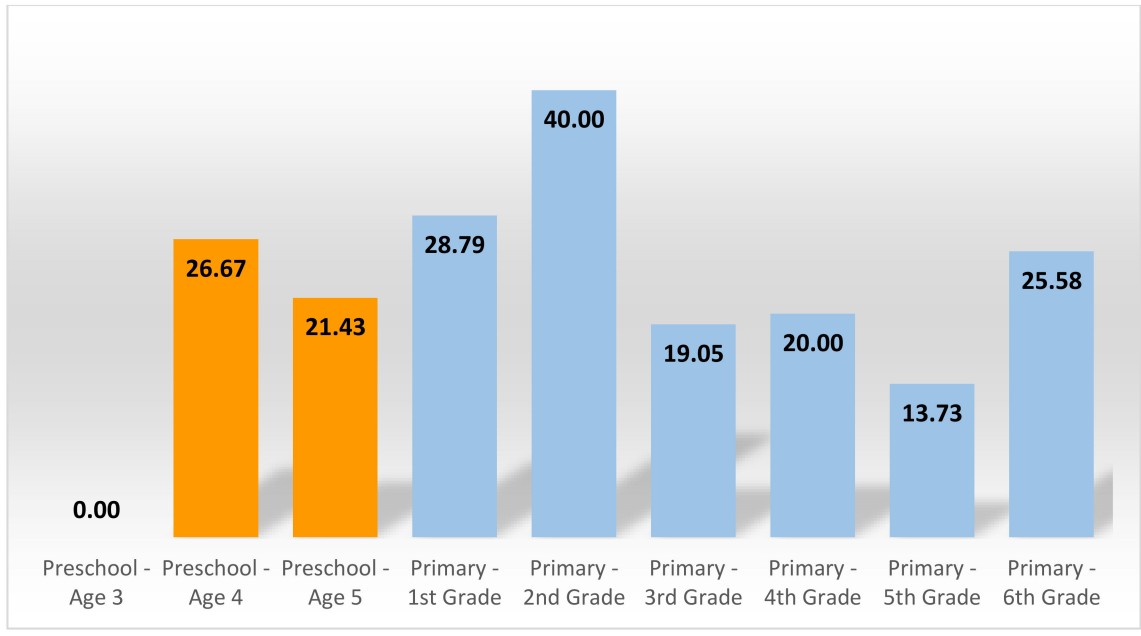

**Figure 10.** Students who did not take school materials with them to lessons (%).

Based on the School Truancy Prevention and Control Programme's official open records, the participating teachers stated that during the 2018/2019 school year, there were 47 children with open records, which represented 11.35% of the school children from Cañada Real Galiana.

Figure 11 shows the distribution of students with a truancy record through the school grades. The number increased in the last year of primary school, to 14% of children from Cañada Real.

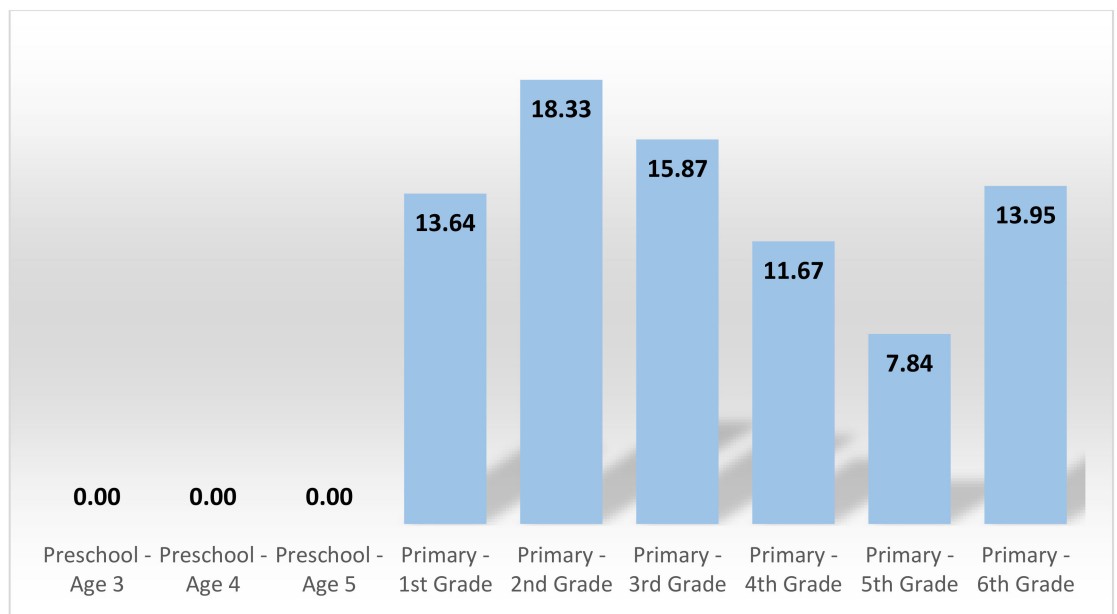

**Figure 11.** Students with a truancy record (%).

In terms of the relationship between Active and Passive Truancy, based on the mentor teachers' perceptions, it can be said that both situations are connected. Those teachers with a greater number of children from Cañada who skipped entire days without a valid excuse, also indicated that they had more students in their classrooms who did not do their homework (Pearson's r = 0.728; $p$ = 0.000) and who lost focus during lessons (Pearson's r = 0.673; $p$ = 0.000). Furthermore, they stated that the different types of Passive Truancy were connected. Mentor teachers with a greater number of students who did not do their homework, also had more children who did not pay attention to lessons (r = 0.712; $p$ = 0.000).

*3.2. Families of Truant Children*

The schools indicated that families of children who live in Cañada Real Galiana had a diverse profile in terms of nationality and ethnicity (Figure 12). The largest group was made up of Spanish Roma families (49.76%), followed by Moroccan (33.33%), Romanian Roma (6.28%), Portuguese Roma (2.90%), Spanish families who were not from a specific ethnic group (4.11%), and other groups of families (4.83%).

The mentor teachers indicated that most of the truant students came primarily from families of Roma ethnicity. The mentor teachers who taught more Spanish Roma students had classes with more children who skipped entire days without a valid excuse (Pearson's r = 0.793; $p$ = 0.000), did not do their homework (Pearson's r = 0.737; $p$ = 0.000), and got distracted from what was happening in the classroom (Pearson's r = 0.634; $p$ = 0.000).

Delving into the mentor teachers' knowledge of the family characteristics of students from Cañada Real Galiana, it is important to highlight that they were unaware of the family situation of 25.85% of these children. As shown in Figure 13, approximately 40% were large families (39.86%) and families who had an extensive family support network (40.34%), whilst there was a scarce presence of single parent families (5.07%).

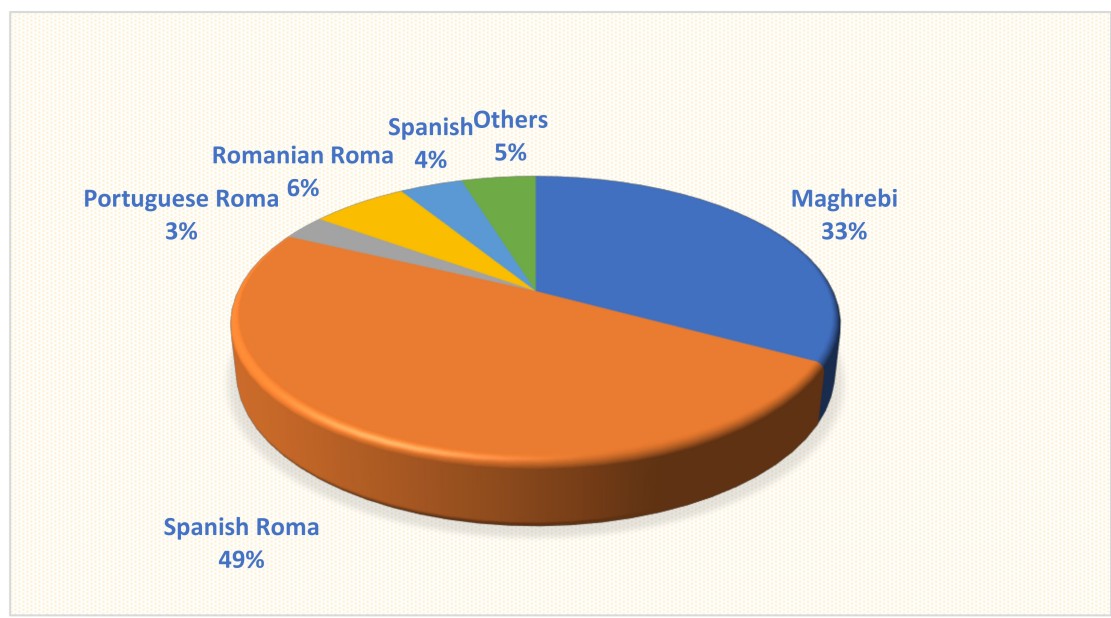

**Figure 12.** Families' ethnic groups.

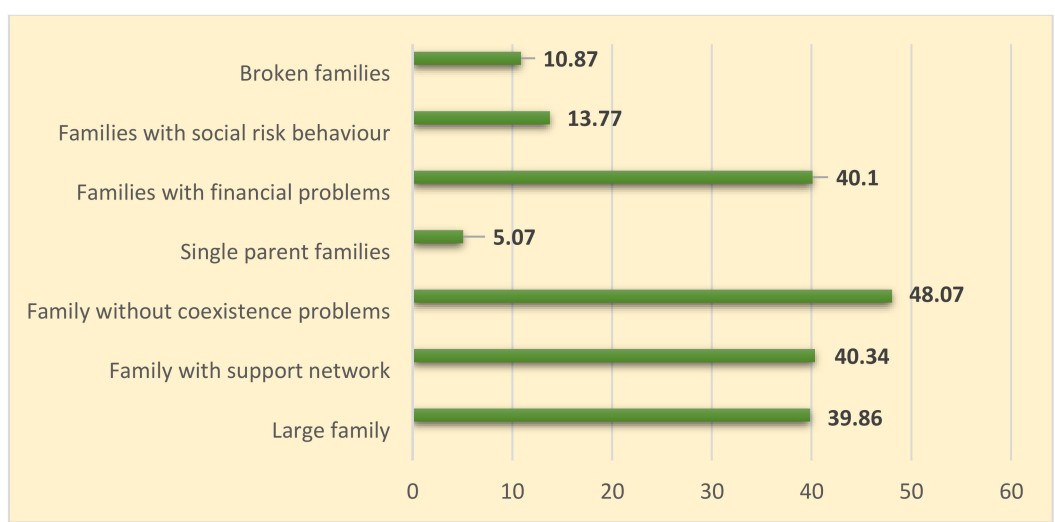

**Figure 13.** Family situation (%).

In terms of the challenges facing these families, based on the information that the mentor teachers had, it can be said that the main problem was financial insecurity (40.10%), followed by other social risk factors such as addictions and domestic violence (13.77%) and family breakdowns (10.39%).

These family difficulties were linked to truancy behaviours amongst these students, with a family's financial difficulties being the most closely linked to both Passive and Active Truancy. Mentor teachers who had more children from financially vulnerable families in their classes also had more children who skipped entire days without a valid excuse (Pearson's r = 0.487; $p$ = 0.000), did not pay attention (Pearson's = 0.444; $p$ = 0.000), did not do their homework (Pearson's r = 0.420; $p$ = 0.000), and did not take their materials with them to school (Pearson's r = 0.407; $p$ = 0.000).

Social risk factors in families (addictions, domestic violence, etc.) were also related to Passive Truancy. When in a classroom there was a high number of students who belonged to these types of families, there was also a higher number of children not bringing to class the sourcebooks or school supplies they needed for the school day (Pearson's r = 0.530; $p$ = 0.000) as well as a higher number

of students who lost focus during lessons (Pearson's r = 0.431; *p* = 0.000). Finally, the link between physically skipping entire school days and belonging to a large family was noteworthy. Mentor teachers with more students from large families had more students who skipped entire school days without a valid excuse (Pearson's r = 0.432; *p* = 0.000).

### 3.3. School Results for Children from Cañada Real Galiana

School truancy is a fundamental problem for any child's successful education. School mentor teachers indicated that children from Cañada Real Galiana showed significant underachievement, with 60% of these children being below the average (Figure 14). It is important to highlight that in the 2018/2019 school year, 40% of these students failed more than four subjects.

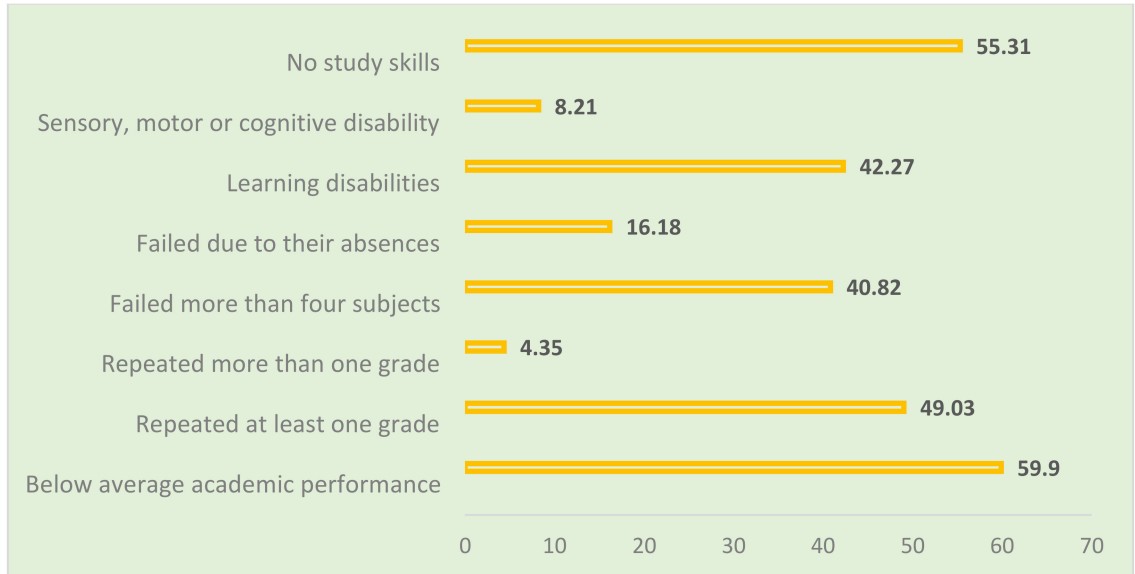

**Figure 14.** School performance (%).

The curricular gap amongst these children was important, as nearly 50% repeated at least one grade. Furthermore, they faced challenges to their educational career in preschool and primary school, as a result of a lack of study skills (55.31%), learning disabilities (42.27%), or some kind of disability (8.21%).

Regarding performance, children from Cañada Real already performed below the class average at preschool, and This became worse throughout primary school. As shown in Figure 15, 4th and 6th grade in primary school were the years where the percentage of students with below-average school performance became more significant. According to their mentor teachers, it reached its flash points (over 70%).

Furthermore, it is important to consider the high number of subjects that these students failed—more than four—throughout all grades of primary school. As shown in Figure 16, this phenomenon was significant in 1st and 6th grade in primary school, where more than half of the students from Cañada Real failed more than four subjects.

From the information provided by the mentor teachers, it can be seen that the curricular gap for children from Cañada Real Galiana was a particular problem for students in 6th grade, where 74.42% of these students repeated a grade (Figure 17) and 27.91% repeated at least two (Figure 18).

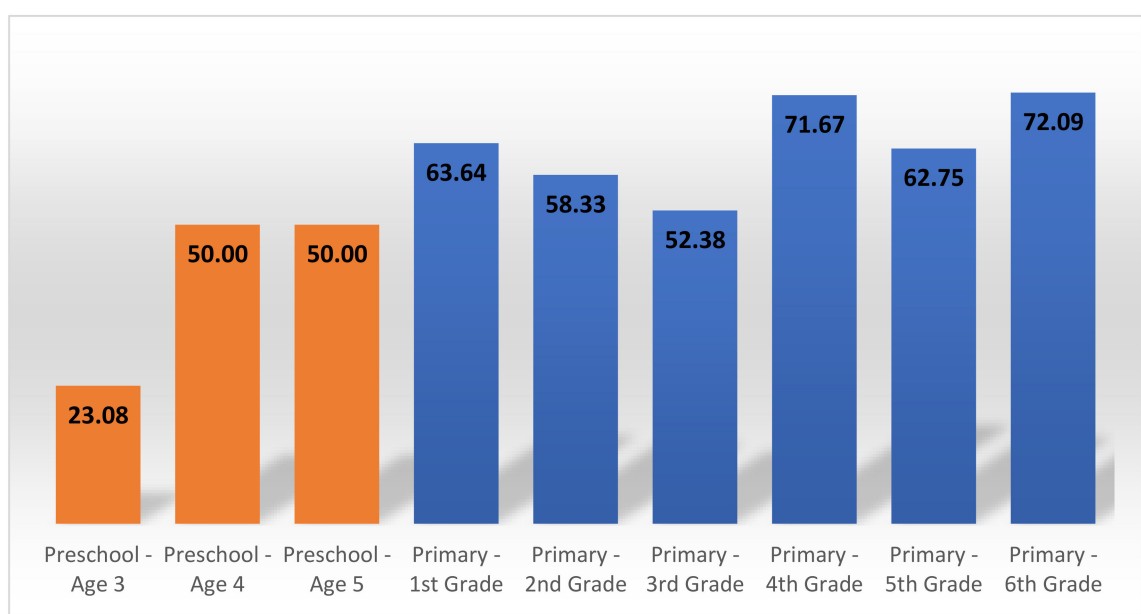

**Figure 15.** Students with performance that was below the class average (%).

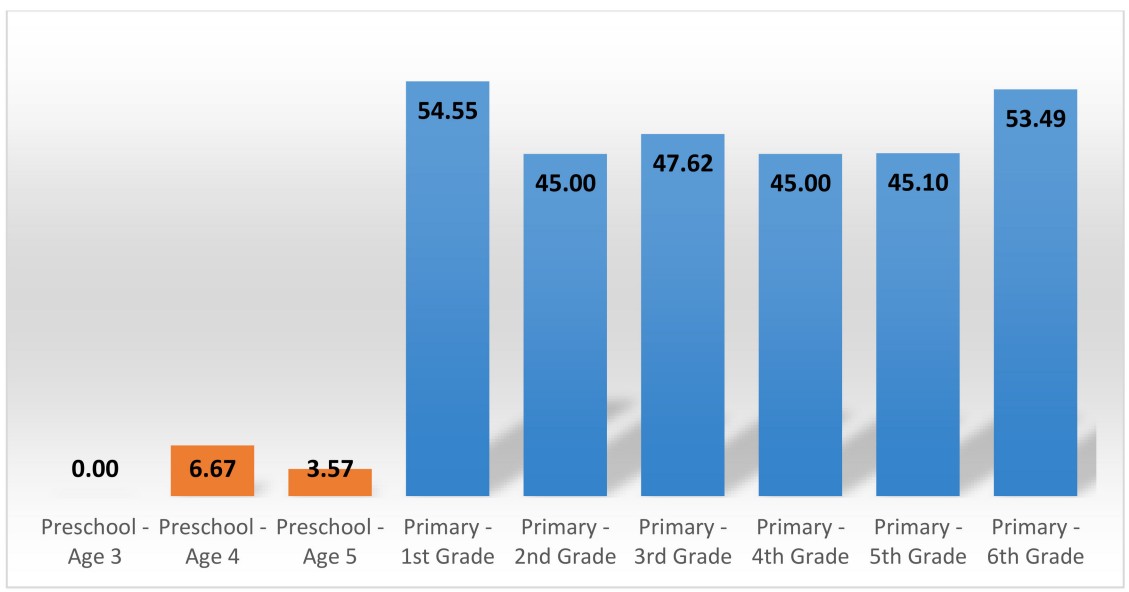

**Figure 16.** Students who failed more than four subjects (%).

Towards the last stage of preschool, more than a quarter of the children had already repeated a grade (28.57%), and from primary 3rd grade more than half of the students in each grade should have been in a higher grade.

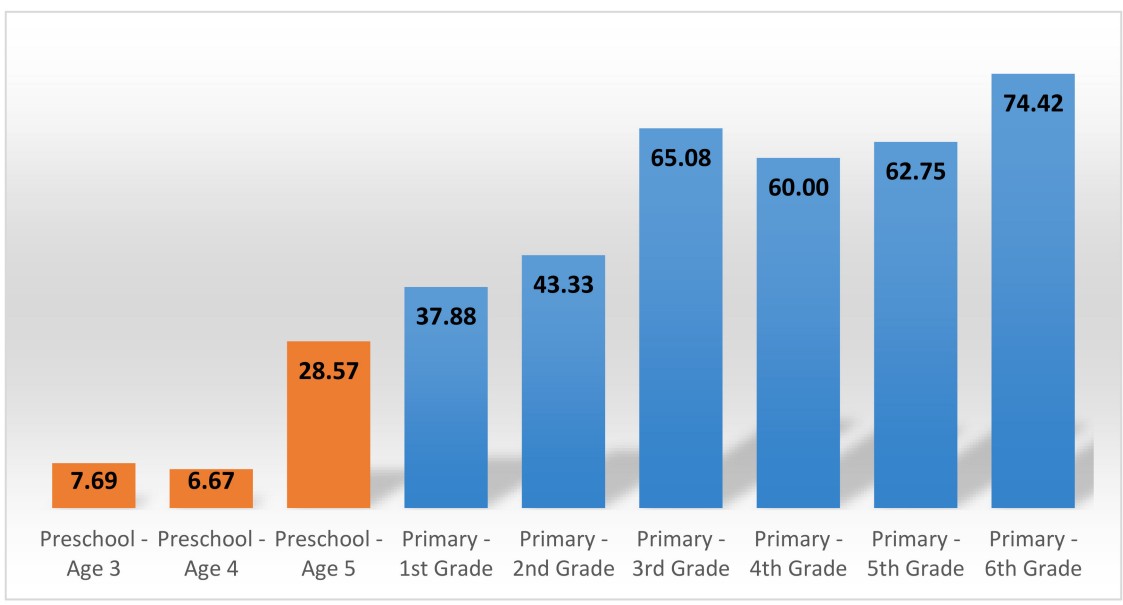

**Figure 17.** Students who repeated a grade (%).

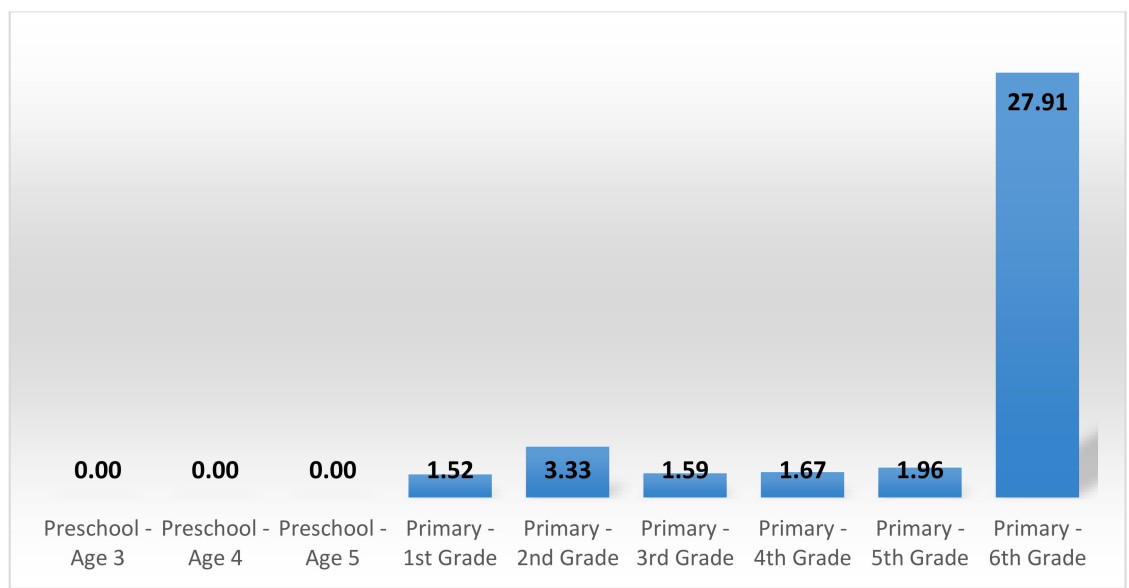

**Figure 18.** Students who repeated more than one grade (%).

The lack of study skills was evident throughout all the school years (Figure 19). According to the mentor teachers, more than half of primary school children from Cañada Real did not have these skills, This becoming especially significant in 6th grade, where it became more apparent as over 70% of the students lacked study skills.

To complete the analysis of these children's academic results, it is important to emphasise the learning disabilities they experienced. According to the mentor teachers, these affected more than a quarter of the students from Cañada Real in the last two years of preschool, further aggravated during primary school years. First and 6th grade were the worst in This respect, with more than half of students in each grade (56.06% and 60.47% respectively) affected, as shown in Figure 20.

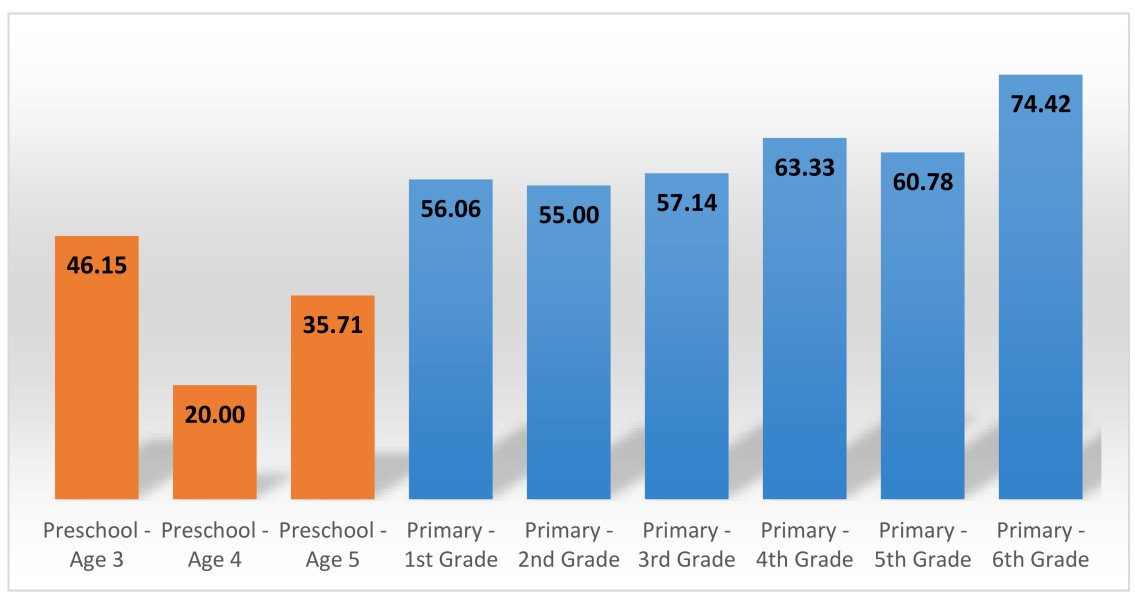

**Figure 19.** Students who lacked study skills (%).

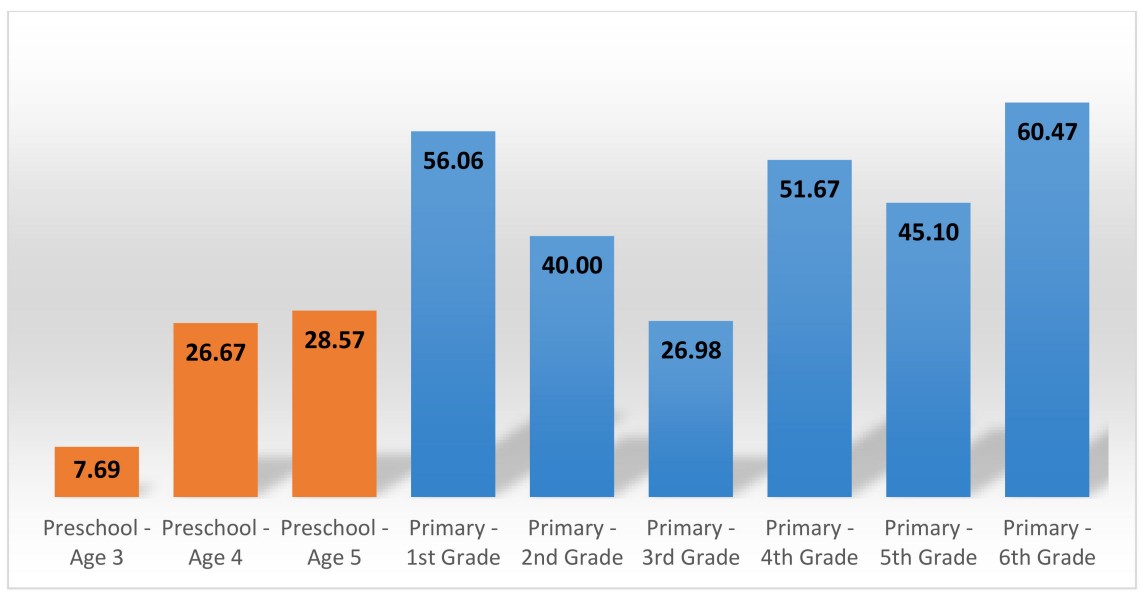

**Figure 20.** Students with learning disabilities (%).

Statistically significant correlations were found amongst students from Cañada Real Galiana between Active Truancy and the challenges they had with their academic results, curricular gaps, and study skills. The teachers who had a greater number of students who skipped entire days without a valid excuse, also claimed to have more children with below-average school performance in the classes (Pearson's r = 0.742, $p$ = 0.000), children who repeated one grade (Pearson's r = 0.687; $p$ = 0.000), and children who did not have study skills (Pearson's r = 0.744; $p$ = 0.000). When there were a greater number of students with open truancy records, there were also more students who failed subjects (Pearson's r = 0.612; $p$ = 0.000) and were in a lower grade than they should have been (Pearson's r = 0.530; $p$ = 0.000).

Finally, it is important to remember that, although children might attend lessons, Passive Truancy is also usually related to challenges to academic progression amongst these students, based on the mentor teachers' perceptions. Those who taught a greater number of students who did not do their homework, ended up with more students with a curricular gap (Pearson's r = 0.686; $p$ = 0.000) with

worse academic results, either because they failed more than four subjects (Pearson's r = 0.664; $p$ = 0.000) or because their overall performance was below the class average (Pearson's r = 0.779; $p$ = 0.000), and they lacked study skills (Pearson's r = 0.813; $p$ = 0.000). Furthermore, the greater the number of children who got distracted during lessons, the more likely they were to re-take a year (Pearson's r = 0.617; $p$ = 0.000), fail a subject (Pearson's r = 0.664; $p$ = 0.000), perform worse than their peers (Pearson's r = 0.758; $p$ = 0.000), and have problems with their study skills (Pearson's r = 0.705; $p$ = 0.000).

## 4. Discussion and Conclusions

School truancy during compulsory education is one of the manifestations of exclusion that children who live in vulnerable areas suffer from. This is demonstrated by This study's results, which show that, although not all children who live in Cañada Real Galiana skip lessons without a valid excuse, the prevalence of Active Truancy is significant, affecting half of these children. This finding is in line with other studies, which show high truancy rates in children of ages similar to the ones contemplated in This study [15,50–54], especially in children of low-income households [6,31,32,43,48].

Skipping school is just the tip of the iceberg when it comes to the unequal educational opportunities that the children from Cañada Real Galiana suffer from. Their educational problems manifest themselves in Passive Truancy behaviours, which occur when the student does not actively participate in the teaching–learning process by not doing their homework and/or losing focus during lessons (40% of children), or not taking materials with them to school (a quarter of the children). In these situations, despite attending school on a regular basis, children can display emotional and motivational dissociation problems that are not only more likely to make them skip school without a valid excuse, but can also lead to early school dropout.

Although preschool is not a compulsory education stage in Spain, in fact, most children enrol in these schools as it provides the first school experience, which is crucial when entering higher stages of education. It is important to emphasise that many children in Cañada Real are not enrolled at preschool, and those who are already show tendencies of Active and Passive Truancy, resulting in them skipping entire school days and not having study skills or paying attention in lessons (at least a quarter of children from Cañada show these behaviours). This finding is consistent with Ready's conclusions [55]. Early truancy can be considered a risk factor for disengaging from school activities in subsequent stages of education, as the academic, behavioural, and routine expectations become greater. When children enter compulsory education at the age of 6, if their preschool experiences have been positive, they enter the new stage having previously learned about routines and skills, whilst also having experienced a connection with their fellow classmates [9]. This puts them in an ideal situation for what they will learn during compulsory primary education, becoming a critical prevention factor for early school truancy at primary school.

School truancy at primary school is a prevalent and important issue. Around a third of children from Cañada Real skip entire days, do not do their homework, or do not pay attention during lessons. Given the compulsory nature of primary education in Spain, This type of truancy is less visible, but it has severe consequences for children as they move up to the higher school grades with a track record of disengagement from school activities. This may lead to students falling into a cycle in which the lack of study skills and attendance become problems more difficult to overcome, and, therefore, make the children more susceptible to early school dropout at the following education stage.

The grades that are the links between educational stages, 1st and 6th grades, are those that have a comparatively higher prevalence of truancy, with 60% of children missing lessons during the entire school day in the last year of primary school. School truancy in these years is a wake-up call for the need to implement care, monitoring, and assistance measures for these children in order to correct the trend and inspire their future educational journey.

Furthermore, from the teachers' perspectives, Passive and Active Truancy reinforce one another, increasing the challenges surrounding these children's progression. There is a link between skipping entire days without a valid excuse, not doing homework, and passive behaviour in the classroom.

There is also a link between Passive Truancy dynamics, making it harder for children who do not take part in lessons to do their homework when they leave school, whilst those who do not keep up to date with their homework find it harder to actively follow the class [3].

The study has also exposed some of the existing vulnerability conditions faced by the people of Cañada Real, which remain closely linked to school truancy amongst children. There is a greater prevalence, both in terms of Passive Truancy as well as Active Truancy, in Spanish children of Roma ethnicity and those who live in financially weak families. The largest cultural-ethnic group in Cañada Real Galiana includes Spanish Roma people, a group that suffers from a lack of educational continuity and success throughout school, as well as high levels of school dropout during secondary education. These findings are consistent with the idea of truancy being the result of the interaction of multiple factors: family, culture and socioeconomic level, which if also found in the literature [24,36,43,56–60].

As a result, it should be emphasised that families who live in poverty are more unlikely to have the necessary resources and skills to compensate for the early educational gaps that have arisen from a fragile educational journey prior to arriving at secondary school. For these students in the poverty gap, the addition of an educational gap increases exclusion and the discomfort they suffer from, determining their future educational opportunities and, ultimately, their personal progression opportunities and social mobility, whilst also reinforcing the intergenerational transmission of poverty. These findings are similar to other results found in the literature [6,36,55].

It can therefore be observed that the children from Cañada Real Galiana are heading towards educational exclusion. From as early as preschool, teachers highlighted these children's poor school performance, which is brought to light by the high number of failed grades, lack of study skills, and learning disabilities that are seen during primary school. All of these aspects contribute to a situation of greater vulnerability when it comes to starting secondary education.

A significant relationship has been discovered between school truancy, school performance, curricular gap, and motivation for studying amongst children from Cañada Real Galiana. Findings are consistent with other results found by different authors [15–23], which are even higher in vulnerable students [61–64]. The mentor teachers who have more children in their classes who skip entire days, do not do their homework, and do not pay attention during class, indicated that they also have more students with academic results below the class average, who repeat grades and lack study skills. Indeed, it is reasonable to think that by missing school and disengaging from educational activities at home and at school, they are unable to maintain their efforts, motivation, and development. Furthermore, their educational challenges, in terms of results and progress against the curriculum, reinforce Active and Passive Truancy. As a result, truant children move through the earlier school grades accumulating gaps in their knowledge, getting poor marks, or repeating a grade. They therefore lose any relationships they have established in previous years and, rather than being a source of reinforcement and personal affirmation, school starts to undermine their self-confidence and educational aspirations.

In conclusion, Cañada Real Galiana is an area that suffers from social exclusion in its most extreme form, and the children who live there are affected from an early age by dynamics that promote educational exclusion and have an impact on their inclusion, enjoyment, and continuity at school, and also set them on a path of early educational abandonment which will limit their future ability to get out of the poverty cycle.

The outcome of the investigation makes us come to conclusions similar to the ones of other authors in terms of the need of taking preventive action on truancy in early grades in school [6,12,15]. Working towards reducing early school truancy amongst the children who live in Cañada Real Galiana is an urgent task for guaranteeing their current and future well-being. It is important to consider not only physical absence from school, but also the children's disengagement from school life and routines, which is worrying. Detecting, preventing, and taking early action against truancy at primary school is critical for avoiding early school dropout and ensuring that school is a real agent for promoting advancement for the children who live in extreme exclusion.

It is necessary to develop simultaneous and coordinated actions in three areas: the family, the school, and the children themselves. Educators must combine both individualised and group attention and the community approach. Also, they should incorporate all the resources of their environment as well as working with associations and entities of the third sector and the community itself. This approach is especially necessary in contexts of maximum social vulnerability.

It is essential to carry out specific work with the families of truant children, who are the people in charge of their attendance at school during the preschool and primary stages. It is necessary to develop actions to raise awareness and make families realise the decisive role of education in the future of their children. Their involvement and active participation in the educational process should be publicised, and they should be supported with all the necessary resources to compensate for their deficiencies and to strengthen their parental and educational capacities.

The school and the mentor teachers have a fundamental role. It is essential to foster the early detection of school truancy and to develop specific detection, prevention, and intervention programmes for preschool and primary stages. Likewise, teachers, especially mentor teachers, of socially disadvantaged children must be supported. Teachers must be trained in inclusive education to meet the needs arising from the social situation of these students. The school and the mentor teachers have to intensify the relationship with the families, improving communication, promoting meetings, and supporting them in their needs of educational support for their children.

The children themselves require specific actions with the objective of increasing their motivation to study. It is necessary to link their study with their future lives and present them with models and references that show them life alternatives different from those around them in their immediate environment. It is essential to detect and prevent performance problems as soon as possible. All the support resources available in the school environment and the community must be used and specific plans adapted to their educational needs must be implemented. These plans should be oriented towards working on learning difficulties and educational support to reduce underachievement rates and the curricular gap and to avoid the repetition of grades. It is very important to intensify the bonding of children with their school, making them feel part of it, not only from an academic point of view, but through their participation in extracurricular activities of leisure, sports, art, etc. These activities allow them to strength their relationships with other students, consolidate their identity, and increase their sense of belonging to the school.

**Supplementary Materials:** The following are available online at http://www.mdpi.com/2071-1050/12/20/8464/s1, Table S1: Study data collection. The authors will provide the research data in SPSS format (.sav) on request.

**Author Contributions:** Conceptualization, S.L. and E.R.; methodology, B.U. and R.M.; formal analysis, B.U. and R.M.; investigation, S.L., B.U., R.M. and E.R.; resources, S.L. and E.R.; writing—original draft preparation, S.L., B.U., R.M. and E.R.; writing—review and editing, S.L., B.U., R.M. and E.R.; project administration, S.L. and B.U.; funding acquisition, S.L. and B.U. All authors have read and agreed to the published version of the manuscript.

**Funding:** This research was funded by Madrid City Hall.

**Conflicts of Interest:** The authors declare no conflict of interest. The funders had no role in the design of the study; in the collection, analyses, or interpretation of data; in the writing of the manuscript, or in the decision to publish the results.

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
