# Peer review of "Primary Education Truancy and School Performance in Social Exclusion Settings: The Case of Students in Cañada Real Galiana"

_sustainability, doi:10.3390/su12208464_

Round 1
Reviewer 1 Report
The Authors presented a paper titled: " Primary education truancy and school performance in social exclusion settings: the case of the students in Canada Real Galiana" that developed an interesting issue from social and ethical point of view. The paper too long in some part (as Introduction) is very well documented, well written and adequate from a methodological as well as statistical analysis.Even if related to a specific geographic area the obtained results have a social and educational impact.
What I ask the Authors is to implement the final part of Discussion and Conclusion. They at the end talk about taking early action against truancy of primary school , crucial aspect. They should try to add some specific action (towards students, towards teachers and towards parents , all indicted in some way in the partially poor results) as example, otherwise the reader remains with empty promises, hypothetical strategies as reported in many other papers and I consider a strong point following the presented data a true suggestion on how to manage the situation to decrease the negative aspects of this behavior.
Author Response
Dear collegue,
Thank you for your revision and your feedback on our parper. We have identified your observations as pertinent and relevant. Following your advice, we have introduced in the Discussion and Conclusions part (from line 608 to 637), our proposal of specific actions on the issue to reduce truancy in Prescholar and Primary School directed towards family, school and teachers and with the children themselves. The main goal of these actions lies in the prevention, identification and intervention in early stages of these truancy-related behaviours. We hope our modifications are aligned with your observations.
Kind regards,
Reviewer 2 Report
The paper present relevant topic for the scientific community.
Author Response
Dear collegue,
Thank you for your revision and your feedback on our parper. With the received observations, we have introduced in the Discussion and Conclusions part (from line 608 to 637), our proposal of specific actions on the issue to reduce truancy in Prescholar and Primary School directed towards family, school and teachers and with the childrem themselves. The main goal of these actions lies in the prevention, identification and intervention in early stages of these truancy-related behaviours. Thank you again for your feedback.
Kind regards,